# Efficacy of Home Oral-Hygiene Protocols during Orthodontic Treatment with Multibrackets and Clear Aligners: Microbiological Analysis with Phase-Contrast Microscope

**DOI:** 10.3390/healthcare10112255

**Published:** 2022-11-10

**Authors:** Paolo Caccianiga, Alessandro Nota, Simona Tecco, Saverio Ceraulo, Gianluigi Caccianiga

**Affiliations:** 1School of Medicine and Surgery, University of Milano-Bicocca, 20900 Monza, Italy; 2Dental School, IRCSS San Raffaele Hospital, Vita-Salute San Raffaele University, 20132 Milan, Italy

**Keywords:** oral hygiene, oral prevention, orthodontics, multibrackets, clear aligners, microbiological analysis, phase-contrast microscope, oral irrigator

## Abstract

The purpose of this study is to analyze the microbiota of patients undergoing orthodontic treatment with multibrackets and transparent aligners. The second goal is to evaluate the effectiveness of the oral irrigator on the oral hygiene and periodontal health of orthodontic patients. Fifty patients (27 F, 23 M; mean age 21.5 years) were recruited for the study, then divided into two groups. Group A underwent fixed orthodontic therapy with multibracket, and a home protocol that included manual orthodontic toothbrush, interdental brushes, and one-tuft brushes. Group B used transparent aligners for 22 h a day and a home protocol that included a manual brush with soft bristles and dental floss. After 3 months, all patients of the two groups, A and B, underwent plaque evaluation with a phase-contrast microscope. If the test result showed non-pathogenic bacterial flora, the subject continued with the traditional home oral-hygiene protocol. If the test detected pathogenic flora, the subject changed the home protocol, with a sonic toothbrush and oral irrigator, while the microbiological analysis continued to be performed after 3 months. After 3 months, 10 out of 25 patients treated with multibrackets (group A) and only 3 out of 25 patients with aligners (group B) passed from non-pathogenic flora to pathogenic flora. After 6 months, using the oral irrigator and a sonic toothbrush for 3 months, all subjects returned to non-pathogenic flora. This study confirms that in patients treated with multibrackets, the risk of developing unfavorable microbiota increases compared to those treated with clear aligners. The use of an oral irrigator combined with the sonic toothbrush seems to be able to restore good oral hygiene in subjects with pathogenic flora and therefore to be effective at reducing the risk of caries and gingivitis in orthodontic patients.

## 1. Introduction

When iatrogenic factors intervene to modify the pre-existing balance of the oral cavity, the balance of the bacterial flora that live in it is also modified, resulting in an increased risk of developing oral diseases such as caries, gingivitis, and periodontal diseases [1,2,3,4].

The introduction of orthodontic appliances, which are increasingly in demand today due to aesthetic needs and a greater awareness of oral health, can lead to imbalances in the oral ecosystem [5,6,7]. In particular, greater plaque accumulation was found in patients with fixed orthodontic appliances and greater colonization by pathogenic bacteria such as Streptococcus mutans and Lactobacilli due to the difficulty in maintaining good oral hygiene [8,9].

For about 50 years, Streptococcus mutans has been considered a key bacterium for initiating the carious process, while Lactobacilli are believed to be responsible for advancing the destruction of dental tissue. The growth of these bacteria is favored by a diet rich in carbohydrates and an increase in the number of retention areas in the mouth, as happens in the presence of orthodontic appliances [8,9].

In recent decades, patients have begun to request to be treated with invisible aligners, which, in addition to the considerable aesthetic advantages compared to traditional orthodontic treatments, seem to respect periodontal tissues more, thanks to the lower accumulation of pathogenic bacterial plaque [3,4]. However, the literature lacks studies comparing bacterial plaque between patients with invisible aligners and patients with multibracket appliances; the only study published in this regard in which a microbiological evaluation by PCR was carried out is the one previously mentioned conducted by Levrini et al. [10], in which a lower bacterial load was found in the oral cavity of patients treated with invisible aligners.

To reduce this increased risk of alteration of the oral balance caused by the introduction of orthodontic equipment, it is essential that the orthodontic patient correctly follows the oral hygiene instructions provided [11,12,13].

However, it has also been seen that the use of orthodontic equipment is able to complicate daily oral hygiene maneuvers, modifying the patient’s oral microbiota and predisposing him to the development of future oral diseases affecting dental or periodontal tissues [5,6,7].

The purpose of this study is to analyze the microbiota of patients undergoing orthodontic treatment with multibrackets and transparent aligners. The second goal is to evaluate the effectiveness of the oral irrigator on oral hygiene and periodontal health of orthodontic patients. These evaluations are possible using the phase-contrast microscope for qualitative microbiological analyses of bacterial plaque.

In other words, the rationale of this study is to verify whether the quality of the bacterial plaque of a periodontally healthy patient, under treatment with fixed multibracket or transparent aligner orthodontic therapy, can be controlled by adopting specific home oral-hygiene protocols.

## 2. Materials and Methods

Fifty subjects (27 females, 23 males; 13–30 years old; mean age 21.5 years) were recruited from a larger group of 124 patients.

The enrollment process was based on the following criteria:Age between 13 and 30 years,Permanent mandibular dentition,Angular class I malocclusion,6–6 mild lower crowding measured on dental model,No diastema or spaces in the lower arch,No ectopic tooth,No extraction required or intraoral or extraoral aids,No previous orthodontic treatment,Silness and Löe Plaque index equal to 0,PSR 0,No gingival recession,Periodontal probing not greater than 3 mm in all the sites examined.

The sample was divided in two groups: Group A underwent metal multibracket self-ligating therapy (American Orthodontics, Sheboygan, WI, USA) [Figure 1a], and group B used clear aligners (Ortobel System, Bergamo, Italy) [Figure 1b]. These patients wore the aligners 22 h a day and replaced them every 14 days. Patients with multibracket self-ligating appliances combined the use of orthodontic brush, interdental brushes, and one-tuft brushes; patients in therapy with aligners used soft brushes and dental flosses. Dental floss was not recommended for patients with multibracket therapy because, due to orthodontic floss, it would have been difficult to use.

At the beginning of the therapy (T_0_) and after 3 months (T_1_), all patients in the two groups, A and B, underwent plaque sampling and evaluation with a phase-contrast microscope. The authors suggested a qualitative evaluation of a subgingival plaque sample with analysis by contrast-phase microscopy (DM500, Leica, Wetzlar, Germany) (Figure 2 and Figure 3).

The subgingival plaque is taken with a periodontal curette inserted in the gingival sulci in the interdental spaces between the molars, both in the upper and the lower jaw, where bacterial plaque is more likely to be present. The plaque is deposited on a slide and irrigated with a drop of physiological solution. A counterslide is then affixed to the slide. Finally, a drop of oil is placed on top of the slide to concentrate the light of the microscope.

A 40× objective is recommended for an eyepiece of 15 (600 magnification).

Thanks to a camera integrated in the microscope, it is possible to view the bacterial plaque samples on the computer and take screenshots (Figure 4 and Figure 5).

In the preparation, it is possible to visualize which bacteria are present, as well as the number and structures of epithelial cells and polymorphonuclear cells found there. It seems simple to make a distinction between

-Non-pathogenic bacterial flora: immobile, similar to the aerial view of the mainland (Figure 4a);-Pathogenic bacterial flora: in which “Streams” or “basins” with mobile bacteria are visualized, mostly composed of spirochetes (including Treponema Denticola) and Trichomonas Tenax, which flow in motion as if dragged by the current. The most consistent similarity is that of the aerial visualization of the Norwegian fjords (Figure 4b).

After the microbiological evaluation at T_1_, if the test result showed non-pathogenic bacterial flora, the subject continued in the traditional home oral-hygiene protocol. If the test found pathogenic flora, the subject changed the hygiene protocol, based on the use of sonic toothbrush with vertical oscillation and oral irrigator (Figure 6), while the microbiological analysis continued to be performed every 28 days.

The authors suggested the following modified home oral-hygiene protocol: sonic brush with a vertical movement (Broxo OraBrush, Santé Parodonte, Geneva, Switzerland), interdental brushes, and oral irrigators (Broxo OraJets, Santé Parodonte, Geneva, Switzeland), at least two times every day (Figure 7, Figure 8 and Figure 9).

After other 3 months (T_2_, 6 months from the beginning of the therapies), another microbiological evaluation was performed on all patients, both on patients with a traditional home oral-hygiene protocol and on those with a modified home oral-hygiene protocol.

### Statistical Analysis

Data about the number of patients who changed to non-pathogenic flora and about the variations in microbiological composition after the application of a sonic toothbrush with an oral irrigator were compared between the groups using a chi-square statistical test. The threshold for statistical significance was set at *p* < 0.05.

## 3. Results

### 3.1. Initial Evaluation: Before Orthodontic Therapy (T_0_)

Before the 50 patients enrolled in this study began orthodontic therapy (T_0_), all 25 patients of group A and all 25 of group B presented non-pathogenic bacterial flora on microbiological analysis with a phase-contrast microscope. The qualitative microbiological analysis under a phase-contrast microscope showed in all cases an immobile bacterial flora, mostly composed of Gram+. It follows that all patients were in good periodontal condition at the start of therapy (Table 1).

### 3.2. Second Evaluation (T_1_)

Three months later (T_1_), after the 25 patients in group A started orthodontic therapy with multibrackets and adopted the home oral-hygiene protocol with an orthodontic brush, interdental brushes, and one-tuft brushes, and the 25 patients in group B started orthodontic therapy with aligners and adopted the oral hygiene protocol at home with soft brushes and dental flosses, 10 out of 25 patients treated with multibrackets (group A) and 3 out of 25 patients with aligners (group B) passed from non-pathogenic bacterial flora to pathogenic flora (mobile flora, composed also of Gram−) (Table 2).

### 3.3. Third Evaluation (T_2_)

Three months later (T_2_), after the 13 patients (10 belonging to group A and 3 to group B) who had had a microbiological analysis with pathogenic flora started using the modified home oral-hygiene protocol with an oral irrigator and a sonic toothbrush, all 50 patients achieved a microbiological outcome of non-pathogenic bacterial flora (Table 3).

The results obtained at the three evaluation times can be summarized in Figure 10.

### 3.4. Statistical Analysis

The chi-square test was performed at T_1_ to verify whether there was a correlation between patients undergoing multibrackets and aligners orthodontic therapy and the presence of incompatible bacterial flora. Since the *p*-value is <0.05, the results obtained are statistically significant (Table 4).

It was not possible to calculate whether there was a statistically significant correlation with the chi-square test between patients undergoing a traditional home oral-hygiene protocol and those undergoing a modified home oral-hygiene protocol since the latter were the patients who were asked to change protocol after a sample of pathogenic bacterial plaque was found during microbiological assessments at T_1_. In fact, in the T_2_ assessment, patients with modified home oral-hygiene protocol were included in the total patient sample.

## 4. Discussion

The phase-contrast microscope uses two principles of geometry (wavelength and amplitude) to create an image of the illuminated cells [14,15]. Methodologically, the next aspects are important as they strongly influence the results of the analysis regarding sample contamination, sampling technique, and sample preparation. The reproducibility of the above-mentioned techniques is high when a large number of parameters are kept constant. Sample analysis gives us some clinically relevant information [14,16]. The direct examination of the sub-gingival dental plaque under a phase-contrast microscope allows you to view the bacterial morphotypes present in the sample and to immediately obtain information on microbial diversity even in the absence of inflammatory clinical signs. It also allows one to evaluate the microbial density and the presence of motile bacteria associated or not with active periodontal lesions.

The effectiveness of phase-contrast microscopy was monitored in order to provide qualitative data of the bacterial flora that integrates the standard periodontal parameters such plaque index, bleeding index, and loss of clinical attachment during the six-monthly recalls of professional hygiene. Quirynen [17] based the results of his research and his convictions precisely on microbiological evaluation through the phase-contrast microscope. Bollen and Quirynen [18], in 1996, published a pilot study that examined the long-term microbiological effects of a “full mouth” disinfection, controlled with phase-contrast microscopy. Yeom [19], in 1997, used the phase-contrast microscope to evaluate the clinical and microbiological effects of the subgingival deposition of bioabsorbable microcapsules loaded at 10% of minocycline (MC) in 15 adult patients with periodontitis. Quirynen [20] used bacteriological tests with a phase-contrast microscope to evaluate the bacterial plaque around the implant surfaces. Acharya [21], in 2012, made a comparison between the efficiency of three different motivational techniques to maintain good oral hygiene during fixed appliance orthodontic treatment. Phase-contrast microscopy, together with the conventional plaque-detection method and the demonstration of the horizontal brushing-washing method, have a lasting effect on the patient. This reduces the need for frequent strengthening sessions of plaque-control programs compared to motivational tests and conventional plaque control measures. A randomized controlled study by Koca et al. [22] investigated the effects of using a phase-contrast video technique on education in oral hygiene training. The participants were blinded to the type of education method. Before orthodontic therapy, the control group was trained only by the conventional method, while the test group was trained by the phase-contrast video-microscopy method in addition to the conventional method. The plaque and gingival index scores and the number of bacteria in test group were statistically lower than those in the control group at the end of the study. They concluded that the training with phase-contrast microscopy has a more positive effect than the traditional method in oral hygiene education.

The analysis of the literature led us to believe that the home oral-hygiene protocols in patients with pathogenic bacterial flora can preferably include the use of the sonic electric toothbrush [23,24,25], thanks to the shock wave generated by the movement of the bristles, combined with the oral irrigator [26,27,28], which is able to disorganize the salivary biofilm, which appears to be the cause of the spread of pathogenic microorganisms in the gingival sulcus [29,30,31].

The present study showed that the use of a specific home oral-hygiene protocol is essential to maintain periodontal health during orthodontic treatment. In fact, oral irrigators are important in order to maintain non-pathogenic bacterial flora in subgingival plaque, as shown by the results of this research. If patients avoid using oral irrigators, in 3 months periodontal conditions deteriorate with an increase in pathogenic bacterial flora detected with phase-contrast microscopy.

It is essential that the patient is always monitored and continuously remotivated to maintain excellent levels of oral hygiene, even if the proposed protocol can seem somewhat demanding both for the patient and for the dental hygienist. The scientific literature proves that a good feasible method to stop the onset and advancement, avoid relapses, and stabilize the results obtained in patients predisposed to periodontal disease is to keep bacteria below the pathological limits using an accurate home protocol.

The scientific assumptions that can lead to the experimentation of a new home protocol for patients in orthodontic treatment are based on the knowledge of the techniques for removing bacterial plaque from the gingival sulcus, and a patient’s own evaluation, through the phase-contrast microscope, of the typing of two specific types of bacterial flora (compatible and not compatible) with the periodontal treatment.

The results obtained show that there is a statistically significant correlation between the risk of developing a pathogenic bacterial flora during orthodontic therapy and the type of therapy: a patient undergoing multibrackets treatment will more easily develop a pathogenic flora than a patient undergoing treatment with clear aligners.

The results also showed the ability of the modified home oral-hygiene protocol with an oral irrigator and a sonic toothbrush to restore a microbiological balance towards a non-pathogenic bacterial flotation, not from a statistically significant point of view but from a qualitative one.

## 5. Conclusions

The present study shows how phase-contrast microscopy could detect sub-gingival plaque modifications due to poor home oral-hygiene protocols application by patients. It also confirms that in patients treated with multibrackets, the risk of developing pathogenic bacterial flora is higher than in those treated with clear aligners, despite the type of home oral-hygiene protocol adopted. Furthermore, the use of an oral irrigator combined with a sonic toothbrush seems to be able to restore good oral hygiene in subjects with pathogenic bacterial flora, both in patients with multibrackets therapy and with aligners, and therefore to be effective at reducing the risk of caries and gingivitis.

The limitation of our study substantially consists of the fact that the microbiological analysis with a phase-contrast microscope is purely qualitative; therefore, we can only distinguish pathogenic from non-pathogenic bacterial plaque. For a quantitative analysis on the bacteria present in the patient’s gingival sulci, it would be necessary to perform real-time PCR tests, which could provide more significant results on the effect of the oral irrigator on the home oral-hygiene protocol and on the difference between bacterial plaque in patients with multibrackets and in those with aligners. However, these tests are to be performed in microbiology laboratories; therefore, they are more expensive and less comfortable than the phase-contrast microscope analysis, which is conveniently performed in the dental office.

## Figures and Tables

**Figure 1 healthcare-10-02255-f001:**
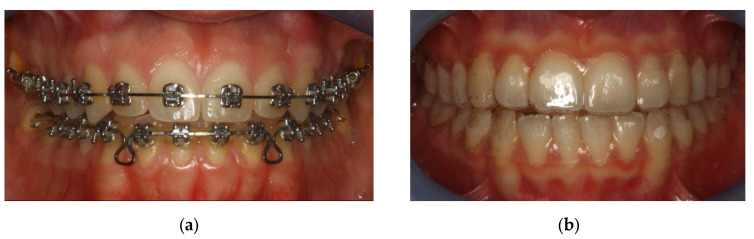
(**a**) Group A: patients with multibrackets. (**b**) Group B: patients with aligners.

**Figure 2 healthcare-10-02255-f002:**
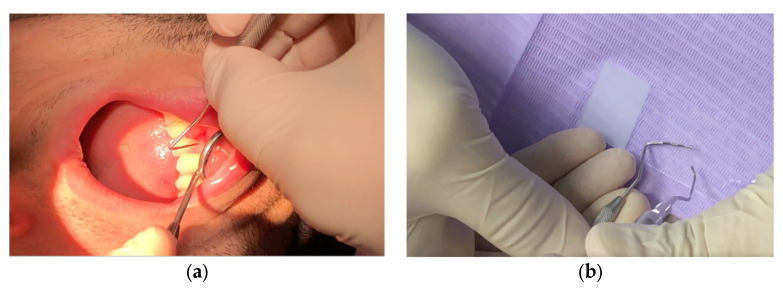
(**a**) Removal of supra and subgingival plaque; (**b**) placement of the plaque on the slide.

**Figure 3 healthcare-10-02255-f003:**
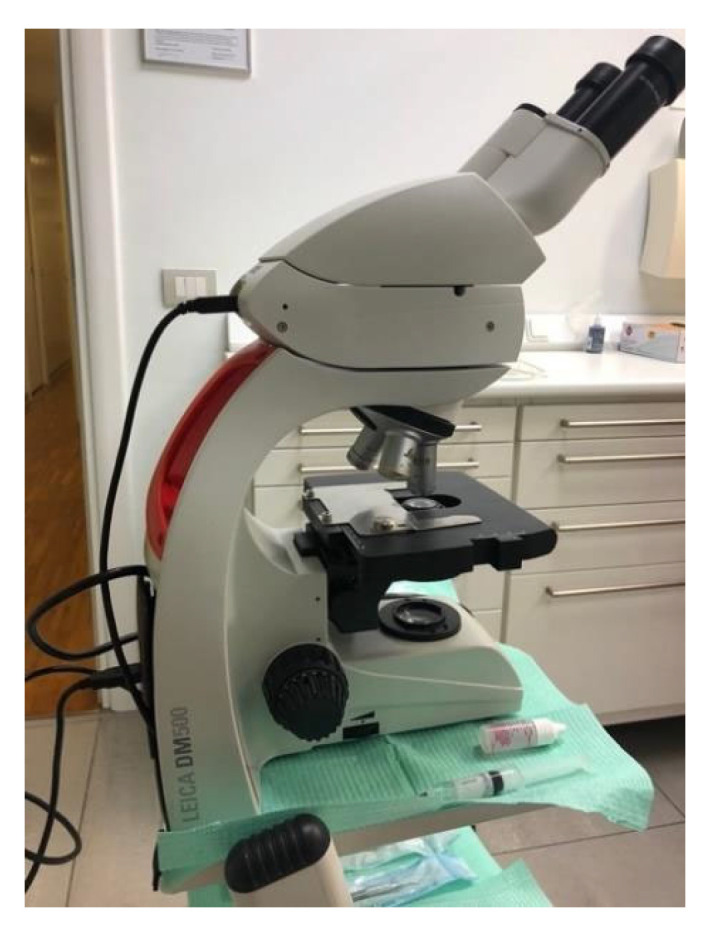
Phase-contrast microscope in periodontal office.

**Figure 4 healthcare-10-02255-f004:**
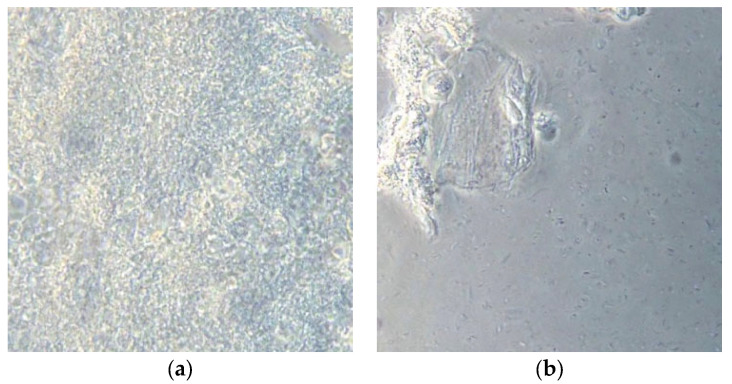
Subgingival plaque images observed under a phase-contrast microscope: (**a**) non-pathogenic bacterial flora (immobile); (**b**) pathogenic bacterial flora (mobile).

**Figure 5 healthcare-10-02255-f005:**
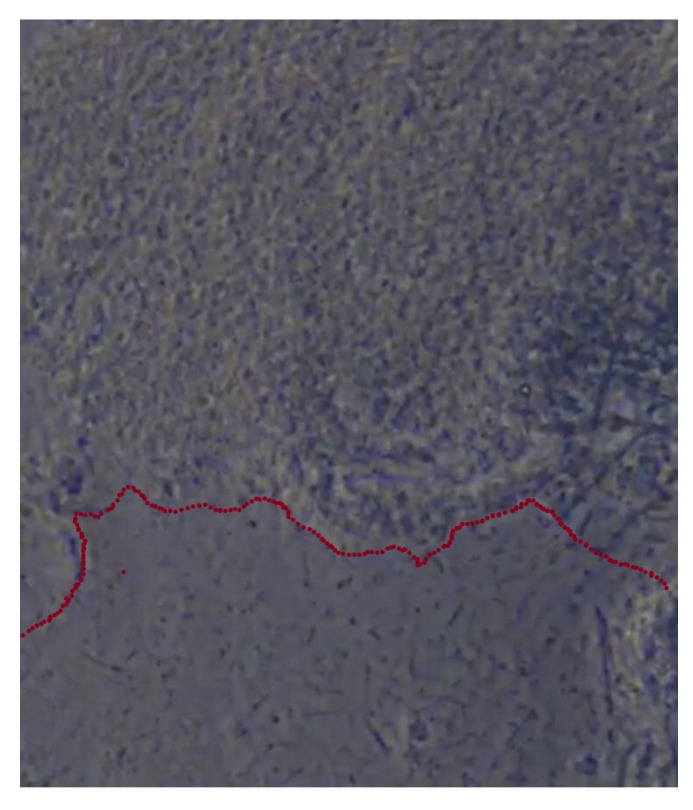
Phase-contrast microscope with dividing line between non-pathogenic bacterial flora (above) and pathogenic bacterial flora (below).

**Figure 6 healthcare-10-02255-f006:**
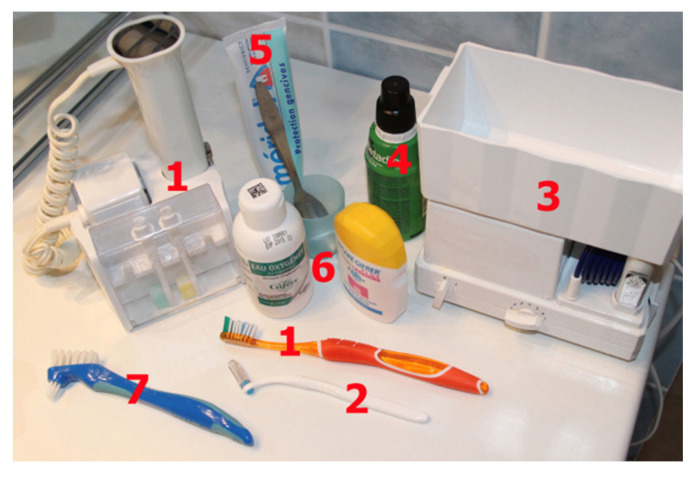
Oral hygiene devices. Seven devices for optimal domiciliary hygiene procedures. The most important are 1, sonic brush with vertical movement, and manual toothbrush; 2, interdental brushes; and 3, oral irrigators.

**Figure 7 healthcare-10-02255-f007:**
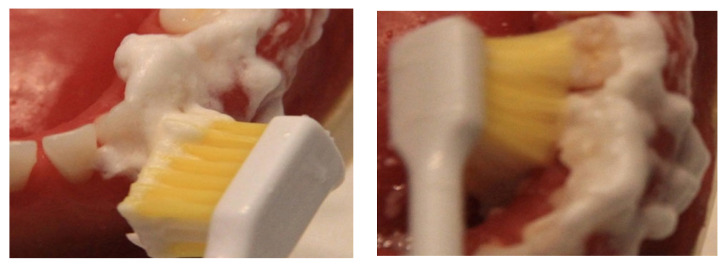
Sonic toothbrush in action.

**Figure 8 healthcare-10-02255-f008:**
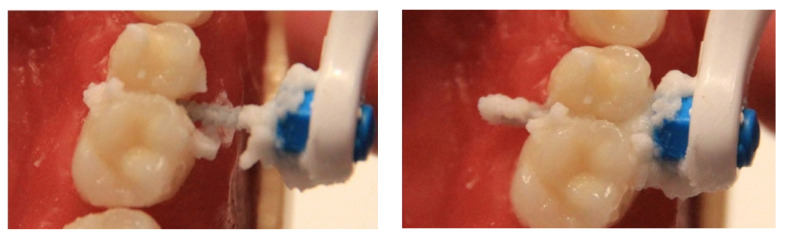
Interdental brushes in action.

**Figure 9 healthcare-10-02255-f009:**
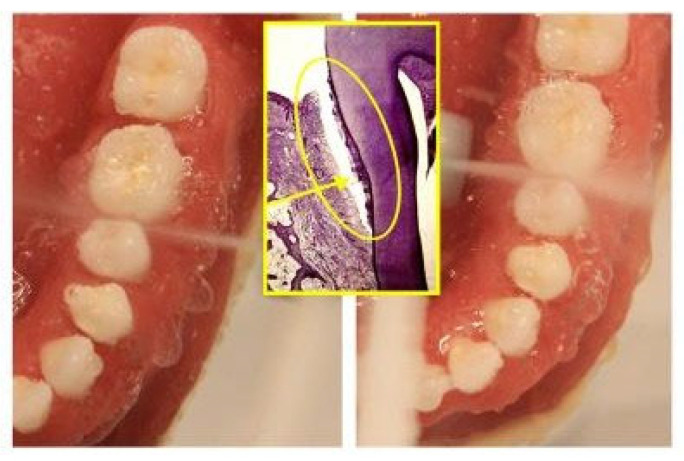
Oral irrigators in action, in order to remove sub-gingival biofilm. (Middle figure: the aim of the oral irrigator is to remove the subgingival plaque, which is deposited in the periodontal pockets. Here, a histological image of a periodontal pocket is shown).

**Figure 10 healthcare-10-02255-f010:**
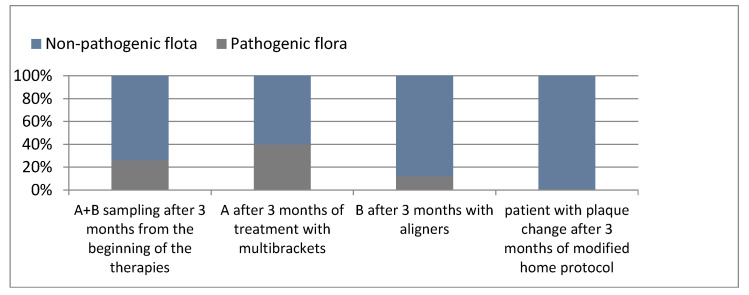
Percentages of patients with pathogenic and non-pathogenic bacterial flora in the various samples.

**Table 1 healthcare-10-02255-t001:** Patients with non-pathogenic and pathogenic bacterial flora at first evaluation (T_0_).

	Non-Pathogenic Flora	Pathogenic Flora
Group A	25	0
Group B	25	0

**Table 2 healthcare-10-02255-t002:** Patients with non-pathogenic and pathogenic bacterial flora at second evaluation (T_1_).

	Non-Pathogenic Flora	Pathogenic Flora
Group A	15	10
Group B	22	3

**Table 3 healthcare-10-02255-t003:** Patients with non-pathogenic and pathogenic bacterial flora at third evaluation (T_2_).

	Non-Pathogenic Flora	Pathogenic Flora
Group A	25	0
Group B	25	0

**Table 4 healthcare-10-02255-t004:** Patients with non-pathogenic and pathogenic flora at first (T_0_), second (T_1_), and third (T_2_) evaluation.

Group	T_0_		T_1_ (3 Months)		T_2_ (6 Months)	
	NPF	PF	NPF	PF	NPF	PF
**A**	25	0	15	10	25	0
**B**	25	0	22	3	25	0
*p-value (Chi-square test)*	*not applicable*		*0.024*		*not applicable*	

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
