# Peer review of "Efficacy of Home Oral-Hygiene Protocols during Orthodontic Treatment with Multibrackets and Clear Aligners: Microbiological Analysis with Phase-Contrast Microscope"

_healthcare, 2022, doi:10.3390/healthcare10112255_

Round 1

Reviewer 1 Report

Dear authors,
in the manuscript the question is original and well defined. The article is properly designed and technically sound. The manuscript is written appropriately and the data which has been analyzed using the highest technical standard. Methods and tools are described in sufficient detail to allow another research to reproduce the experiment and analyses.
However, the English language needs to be improved and some paragraphs need to be revised and made more fluent and clearer for the reader. In the discussion it would be appropriate to provide further details on phase contrast microscopy and on the reliability of its use in other studies as well.
Best regards

Author Response

Dear reviewer,

Thank you for your kind review. We have made an extensive revision of the manuscript according yo yours and other reviewers' suggestions:

We have revised English, re-writing some sentences that were difficult to understand and we have expanded the discussion about phase contrast microscope.

We hope that the manuscript in the current form meets your expectations and can be published in Healthcare.

Faithfully,

Dr. Paolo Caccianiga

Reviewer 2 Report

The paper is short in words and length, work on it.

Abstract:

the abstract must be non-structured as per guidelines.

the conclusion better is not precise, please mention the main achievements of the study as per the objective.

the recommendation better if included in the discussion section.

Introduction:

mention the rationale or justification of the study in 2nd last paragraph.

methods:

Figures 2 and 4... others if any need further description... tag the main finding within the figures with arrow etc.

statistical analysis please include it in the last paragraph of the methods section, it is not to be mentioned in the results.

the results and discussion section please expand with complementary research, and describe the qualitative and quantitative analysis in detailed paragraphs.

Author Response

Dear reviewer,

Thank you for your kind review. We have made an extensive revision of the manuscript according yo yours and other reviewers' suggestions:

  • We have extended the manuscript;
  • Abstract is now non-constructed;
  • The rationale of the study was underlined in Introduction;
  • Figures about bacterial flora in our opinion does not need further elements of clarification because because we performed a qualitative analysis of bacterial plaque with a dichotomous "either pathogenic or non-pathogenic" index based on the "macrostructure" of the plaque and the mobility of the bacteria;
  • Statistical analysis were better underlined in the Methods;
  • Conclusions are now more precise.

We hope that the manuscript in the current form meets your expectations and can be published in Healthcare.

Faithfully,

Dr. Paolo Caccianiga

Reviewer 3 Report

An interesting article about the oral hygiene during orthodontic treatment. However, some issues came to my notice.

1) The manuscript should be evaluated by a native speaker there are a lot of grammar mistakes.

2) Abstract line 12 - change subjects to patients

3) Abstract line 13 - the word and is repeating

4)Introduction Line 35- re-write sentence, what the authors mean with the word change

5)Line 46 - replace germs with bacteria

6)Lines 49-55 re-write the paragraph, it does not make sense

7)Line 64-67 re-write paragraph, it does not make sense, insert null hypothesis

8)M&M: Lines 93-95 Give more info about the method you used for bacteria evaluation. How is it possible to evaluate the pathogen with a microscope only.

9)Line108-114 You should use an additional method for the bacteria

10) Discussion Update the reference you use in paragraphs Line 182-185, 188-191

11) Lines 195-200, 205-206 re-write the paragraph many grammar errors

Author Response

Dear reviewer,

Thank you for your kind review. We have made an extensive revision of the manuscript according yo yours and other reviewers' suggestions:

We have revised English where you suggested, re-writing some sentences that were difficult to understand and we have expanded the discussion about phase contrast microscope.

We hope that the manuscript in the current form meets your expectations and can be published in Healthcare.

Faithfully,

Dr. Paolo Caccianiga

Reviewer 4 Report

Dear Authors,

there are some big questions regarding methodology.

1. How long patients used brackets? What type of brackets they had? Only metal?

2. From which place the dental plaque (biofilm) was taken? Upper or lower jaw? Which teeth? Mesial or distal surface?

3. What about age differences? There is so big range between patiens 13-30 years! Oral biofilm is different in this subpopulations.

4. Which plaque index was used? Please desribe it.

5. What about instructions patients before the study? It was performed or not?

6. Please describe in detailes microscopic picture and show differences between non-pathologic and pathologic microflora.

7. Figure 7 - Is it the own picture?

8. Why authors didn't recommend to "brackets patients" to use dental floss

Introduction - information about adherence of bacteria to different surface is missing - tooth surfaca, brackets, alligners etc.

Discussion - growing/mature of biofilm in the period of time and location is important to discuss and in relation to own results.

Author Response

Dear reviewer,

Thank you for your kind review. We have made an extensive revision of the manuscript according yo yours and other reviewers' suggestions:

  1. Patients at T0 have not yet started orthodontic therapy with multibrackets or aligners, at T1 they have been on therapy for 3 months and at T2 for 6 months.
  2. We specified in the methods where the bacterial plaque was taken;
  3. We have kept a fairly wide range (13-30 years) for the enrolled patients to be able to have an adequate sample, not considering the differences in plaque due to age, which in any case are of little interest to us because we performed a qualitative analysis of bacterial plaque with a dichotomous "either pathogenic or non-pathogenic" index based on the "macrostructure" of the plaque and the mobility of the bacteria;
  4. I have specified that we used Silness & Loe Plaque index;
  5. Patients received no special instruction on oral home hygiene protocol prior to the start of the study. Instructions were given to T0 and T1 for some as described in the methods;
  6. Figures about bacterial flora in our opinion does not need further elements of clarification, for the reason explained at point 3;
  7. I have added explanation about the middle figure, which I think is the cause of your question;
  8. I have addes explanation why we did not suggest dental floss to "bracket patients".

Introduction: we did not look for and add information on bacterial adhesion to surfaces because for this study only the bacterial plaque found in the gingival sulci is of interest.

Discussion: We did not take into account the maturation of the bacterial biofilm for the reasons expressed in point 3.

Discussion: 

We hope that the manuscript in the current form meets your expectations and can be published in Healthcare.

Faithfully,

Dr. Paolo Caccianiga

Round 2

Reviewer 2 Report

The authors team have improved the paper as per suggestions, however grammatical errors are many, which needs consideration further.

Author Response

Dear reviewer,

Thank you for your kind review and for considering our paper fit to be published in Healthcare.

We have further corrected the grammatical errors. If there were others, however, the assistant editors would provide for the correction of the English after the acceptance of the manuscript.

Faithfully,

Dr. Paolo Caccianiga

Reviewer 3 Report

Nice edit

Author Response

Dear reviewer,

Thank you for your kind review and for considering our paper fit to be published in Healthcare.

Faithfully,

Dr. Paolo Caccianiga

Reviewer 4 Report

Please add in the end of the article the limitation of the study.

Author Response

Dear reviewer,

Thank you for your kind review and for considering our paper fit to be published in Healthcare.

We have added the limitations of the study in the conclusions.

Faithfully,

Dr. Paolo Caccianiga